# Contribution of malaria and sickle cell disease to anaemia among children aged 6–59 months in Nigeria: a cross-sectional study using data from the 2018 Demographic and Health Survey

Dennis L Chao [ID],[1] Assaf P Oron,[2] Guillaume Chabot-Couture,[1] Alayo Sopekan,[3] Uche Nnebe-Agumadu [ID],[4] Imelda Bates [ID],[5] Frédéric B Piel [ID],[6] Obiageli Nnodu [ID][7]

For numbered affiliations see end of article.

**Correspondence to**
Dr Dennis L Chao;
dennis.chao@gatesfoundation.org

## ABSTRACT

**Objectives** To estimate the fraction of anaemia attributable to malaria and sickle cell disease (SCD) among children aged 6–59 months in Nigeria.

**Design** Cross-sectional analysis of data from Nigeria's 2018 Demographic and Health Survey (DHS).

**Setting** Nigeria.

**Participants** 11 536 children aged 6–59 months from randomly selected households were eligible for participation, of whom 11 142 had complete and valid biomarker data required for this analysis. Maternal education data were available from 10 305 of these children.

**Primary outcome measure** Haemoglobin concentration.

**Results** We found that 70.6% (95% CI: 62.7% to 78.5%) of severe anaemia was attributable to malaria compared with 12.4% (95% CI: 11.1% to 13.7%) of mild-to-severe and 29.6% (95% CI: 29.6% to 31.8%) of moderate-to-severe anaemia and that SCD contributed 0.6% (95% CI: 0.4% to 0.9%), 1.3% (95% CI: 1.0% to 1.7%) and 10.6% (95% CI: 6.7% to 14.9%) mild-to-severe, moderate-to-severe and severe anaemia, respectively. Sickle trait was protective against anaemia and was associated with higher haemoglobin concentration compared with children with normal haemoglobin (HbAA) among malaria-positive but not malaria-negative children.

**Conclusions** This approach used offers a new tool to estimate the contribution of malaria to anaemia in many settings using widely available DHS data. The fraction of anaemia among young children in Nigeria attributable to malaria and SCD is higher at more severe levels of anaemia. Prevention of malaria and SCD and timely treatment of affected individuals would reduce cases of severe anaemia.

## STRENGTHS AND LIMITATIONS OF THIS STUDY

⇒ This study uses individual-level data from a large, nationally representative survey that includes haemoglobin concentration, malaria testing and sickle cell testing.

⇒ The relatively simple approach used here could be applied to Demographic and Health Survey and similar surveys.

⇒ The data do not include measures of iron or inflammation, which would be needed to identify iron deficiency, a major cause of anaemia.

⇒ We treated malaria rapid diagnostic test results, which are imperfect, as a proxy for malaria infection.

## INTRODUCTION

Anaemia is a major cause of morbidity and mortality among children. According to Global Burden of Disease (GBD) models (http://ghdx.healthdata.org/gbd-results-tool), there were 8.3 million (95% CI: 5.5 to 12.1 million) years lived with disability worldwide among children under 5 years old due to anaemia in 2019, the majority of which (7.8 M, 95% CI: 5.2 to 11.5 million) were due to moderate or severe anaemia. The highest rates of anaemia (adjusted haemoglobin concentration <110 g/L) and severe anaemia (<70 g/L) among preschool-aged children are reported from sub-Saharan Africa (SSA).[1 2]

Anaemia can have many causes, such as nutritional deficiencies, infections and haemoglobinopathies. In low-to-middle income countries (LMICs), children may have several concurrent causes for their anaemia, each requiring a different treatment strategy, for example iron supplementation, chemotherapy, transfusion, antimalarials or deworming.[3 4] Iron deficiency is one of the largest causes of anaemia,[2] but studies estimating the proportion of anaemia due to iron deficiency have highly variable results partly because diagnosing iron deficiency is difficult. A meta-analysis of 23 national

surveys found that 25% of anaemia among preschoolers is associated with iron deficiency with high heterogeneity (range: 0.3%–51%).[5] The gold standard test for diagnosis of iron deficiency, staining of a bone marrow sample for iron, is impractical as a screening method. Serum ferritin is commonly used as a marker of body iron stores, but it is an acute phase protein and therefore increases in the presence of inflammation, which is common in people living in LMICs.[6]

Nigeria is Africa's most populous country and experiences a high childhood anaemia burden. Previously, 60% of Nigeria's moderate and severe anaemia among children under 5 years old has been attributed to iron deficiency and only 12% and 2% to malaria and sickle cell disorders, respectively (http://ghdx.healthdata.org/gbd-results-tool). Because most studies of anaemia are based on relatively small populations, national and regional estimates often need to extrapolate data from these geographically scattered studies to cover the rest of the population. Here, we have explored how Nigeria's 2018 Demographic and Health Survey (DHS), a large, and uniquely detailed data source, could shed light on the contribution of *Plasmodium falciparum* (*P.f.*) malaria and sickle cell disease (SCD) to anaemia. For the first time in DHS history, this Nigerian survey not only tested a population-representative sample of children for malaria infection, but also tested them for haemoglobin subunit beta (HBB) type, including the alleles associated with sickle cell trait and disease.[7] These biomarkers, plus the demographic information obtained from household interviews provide data on over 11 000 children aged 6–59 months. By modelling the risk of anaemia associated with these conditions, while adjusting for key demographic (age, sex) and socioeconomic (household wealth, urban *vs* rural residence, maternal education) risk factors, we can provide a better estimate of the contribution of malaria and SCD to anaemia and explore the inter-relationships between SCD and trait, malaria and anaemia.

## DATA AND METHODS
### Data
We obtained data from Nigeria's 2018 DHS.[7] A sample of 42 000 households were selected for participation in the survey. DHS interviews were conducted from 14 August to 29 December 2018. One-third of households (14 000) were randomly selected for biomarker evaluation. Children 6–59 months old who were de facto residents of these households (ie, slept in household last night) were eligible for biomarker testing (11 536 eligible children). Blood samples taken from a finger or heel prick were analysed with a HemoCue analyzer in the field to determine haemoglobin concentration, which was adjusted for altitude, resulting in 11 206 (97.1% of eligible children) valid haemoglobin concentration measurements. The same finger or heel prick was used to evaluate malaria infection using SD Bioline AG *P.f.* (HRP-II) malaria rapid

diagnostic test (RDT), yielding 11 173 (96.9% of eligible children) RDT results. The same finger or heel prick was also used to evaluate HBB type using SickleSCAN point-of-care test from BioMedomics, yielding 11 186 results (97.0% of eligible children). This test determined the presence of the HbS and HbC alleles, allowing us to determine which children had normal HBB (HbAA), SCD (HbSS or HbSC), sickle cell trait (HbAS) or haemoglobin C trait (HbAC). The nine children who had a recorded HBB type of 'other' were excluded. In total, 11 142 children (96.6% of eligible children) with all three biomarker measurements were included in our analyses. Maternal education was not available for all of these children, so analyses that included maternal education had 10 305 (92.5%) of the children with all three biomarkers. The survey was approved by the National Health Research Ethics Committee of Nigeria and the ICF Institutional Review Board.

The DHS includes sample weights based on the survey design and on household response rates.[7] Applying the weights, the 11 142 children in our analysis represent 11 315.9 children. We applied sample weights for tabulating the number of children and for fitting regression models.

## METHODS
We studied the relationships between anaemia and other factors using three thresholds for adjusted haemoglobin concentration: <110 g/L (anaemia), <100 g/L (moderate-to-severe anaemia) and <70 g/L (severe anaemia).[8]

All the demographic and socioeconomic data came from the DHS. To compare demographic characteristics between children with and without moderate-to-severe anaemia, we applied the Wilcoxon rank-sum test to determine if the distribution of ages was significantly different and a proportions test with $\chi^2$ approximation for the other characteristics (sex, malaria RDT positivity, HbAA, HbAS, HbAC, HbSS, HbSC, rural residence, household in top wealth quintiles and advanced maternal education).

We fitted multivariate quasibinomial logistic and multivariate linear generalised linear models to predict anaemia status and haemoglobin concentration, respectively. We used the svyglm package to fit these models using survey weights (https://r-survey.r-forge.r-project.org/survey/html/svyglm.html). The primary sampling unit is the cluster (eg, a village) and the strata are the DHS-defined strata (the states of Nigeria divided into urban and rural components). When there were no observations of one of the outcomes (eg, anaemia or not anaemic) in a category (eg, children with HbSS), we do not report the odds of the outcome. This effect did not have an impact on the model estimates for the other categories. To compute the effect of sickle cell status among RDT-positive children, we fit models using RDT-positive children with HbAA as the reference group rather than RDT-negative children.

We estimated the adjusted fraction of anaemia attributable to malaria by estimating the number of children with anaemia if no one had malaria using the fitted

logistic regression model described above.[9] To construct the counterfactual simulation where no child is positive for malaria, we used the multivariate logistic regression models fitted to the original data to predict anaemia outcome for eligible children in the DHS after setting all of their malaria RDT results to negative. We interpreted the sum of the probabilities of these children being anaemic to be the number anaemic in the absence of malaria. Among children who were originally RDT positive, the sum of probabilities in the counterfactual population is interpreted to be the number of RDT-positive children who have anaemia that is not malaria-associated. We estimated the fraction attributable to SCD (HbSS/HbSC) by predicting the anaemia status of these children as if they had HbAA.

We computed the 95% CIs of attributable fractions using the bootstrap. For each stratum in the DHS data, we drew clusters with replacement. Sampling by cluster instead of by individual preserved the clustering of children within households and households within villages.[10 11] We performed this procedure 1000 times to generate 1000 populations. We fitted models to these populations to compute the distributions of attributable fractions.

All analyses were performed in R V.4.0.5.

## RESULTS
### Prevalence of anaemia
Among children aged 6–59 months (N=11 142 or 11 315.9 with sample weights), 67.9% were anaemic (adjusted haemoglobin concentration <110 g/L), 41.1% had moderate or severe anaemia (<100 g/L) and 3.0% had severe anaemia (<70 g/L) (table 1). Children who were malaria RDT positive (RDT+) had higher prevalence of severe and moderate-to-severe anaemia than those who were negative (table 1). Of the children with severe anaemia, 80.7% were RDT+, 10.8% had HbSS and 87.8% were either RDT+ or had HbSS (table 1). Among RDT+ children, 7.2% of those with HbAA and 3.8% of those with HbAS had severe anaemia (table 1). Severe anaemia increased sharply with age among those with HbSS (online supplemental figure S1). RDT positivity increased with age (online supplemental figure S2).

Children with moderate-to-severe anaemia were significantly younger and had lower proportions of females, malaria RDT+, HbSS, rural residence, lower three wealth quintiles and lower maternal education compared with children with mild-to-no anaemia (table 2).

### Predictors of anaemia
Malaria RDT positivity, HBB type, age and wealth quintile were significantly associated with all degrees of anaemia (table 3). Among children with HbAA, malaria positivity was associated with increased risk of mild-to-severe (4.05, 95% CI: 3.49 to 4.71), moderate-to-severe (5.06, 95% CI: 4.51 to 5.68) and severe (10.96, 95% CI: 6.96 to 17.25) anaemia (table 3). Among malaria-negative children, HbSS was associated with substantially increased anaemia risk compared with HbAA: an OR of 69.66 (95% CI: 9.63 to 503.87) for mild-to-severe, 22.93 (95% CI: 10.56 to 49.78) for moderate-to-severe anaemia and 76.14 (95% CI: 32.71 to 177.26) for severe anaemia. We were unable to estimate the risk of mild-to-severe anaemia among RDT+ children with HbSS or HbSS because all these children had

**Table 1** Anaemia severity by malaria RDT positivity and sickle cell result

| | | Moderate-to-severe anaemia (<100 g/L) | | Mild or no anaemia (≥100 g/L) | | |
|---|---|---|---|---|---|---|
| | | <70 g/L (severe) | 70–99 g/L (moderate) | 100–109 g/L (mild) | ≥110 g/L (not anaemic) | Total |
| RDT– | HbAA | 35.3 (0.6%) | 1441.8 (25.9%) | 1701.4 (30.6%) | 2386.0 (42.9%) | 5564.4 (100%) |
| | HbAS | 6.1 (0.4%) | 467.9 (32.4%) | 420.4 (29.1%) | 549.4 (38.1%) | 1443.9 (100%) |
| | HbAC | 0.0 (0.0%) | 38.1 (36.1%) | 30.2 (28.7%) | 37.2 (35.3%) | 105.6 (100%) |
| | HbSS | 24.4 (32.3%) | 43.3 (57.4%) | 7.0 (9.3%) | 0.7 (1.0%) | 75.5 (100%) |
| | HbSC | 0.0 (0.0%) | 13.8 (42.4%) | 16.3 (49.8%) | 2.6 (7.8%) | 32.6 (100%) |
| RDT+ | HbAA | 228.5 (7.2%) | 1794.3 (56.5%) | 655.9 (20.7%) | 497.5 (15.7%) | 3176.2 (100%) |
| | HbAS | 30.4 (3.8%) | 442.1 (55.6%) | 183.3 (23.0%) | 139.9 (17.6%) | 795.7 (100%) |
| | HbAC | 0.7 (0.9%) | 49.3 (61.9%) | 14.1 (17.7%) | 15.5 (19.5%) | 79.7 (100%) |
| | HbSS | 12.5 (51.0%) | 10.9 (44.4%) | 1.1 (4.6%) | 0.0 (0.0%) | 24.5 (100%) |
| | HbSC | 2.7 (15.3%) | 12.7 (71.0%) | 2.5 (13.7%) | 0.0 (0.0%) | 17.9 (100%) |
| Total | | 340.6 (3.0%) | 4314.3 (38.1%) | 3032.1 (26.8%) | 3628.8 (32.1%) | 11 315.9 |

The number of children aged 6–59 months, accounting for survey sample weights, is shown for each combination of RDT status, sickle cell genotype and adjusted haemoglobin concentration. In each row, the per cent of children in each anaemia severity category is shown in parentheses.

HbAA, normal HBB; HbAC, haemoglobin C trait; HbAS, sickle cell trait; HbSC, heterozygous for HbS and HbC; HbSS, sickle cell anemia; RDT, rapid diagnostic test.

**Table 2** Characteristics of children with mild-to-severe, moderate-to-severe and severe anaemia

| | Mild-to-severe anaemia | | | Moderate-to-severe anaemia | | | Severe anaemia | | |
|---|---|---|---|---|---|---|---|---|---|
| | <110 g/L | ≥110 g/L | P value | <100 g/L | ≥100 g/L | P value | <70 g/L | ≥70 g/L | P value* |
| N (unweighted) | 7652 | 3490 | | 4660 | 6482 | --- | 350 | 10 792 | |
| Mean age in years | 2.5 (SD: 1.3) | 3.0 (SD: 1.2) | <0.001 | 2.5 (SD: 1.3) | 2.8 (SD: 1.3) | <0.001 | 2.4 (SD: 1.3) | 2.7 (SD: 1.3) | <0.001 |
| % female | 48.1% | 52.4% | <0.001 | 46.4% | 51.6% | <0.001 | 45.4% | 49.6% | 0.14 |
| % RDT+ | 46.6% | 19.1% | <0.001 | 57.5% | 24.0% | <0.001 | 82.3% | 36.5% | <0.001 |
| % with HbAA | 76.7% | 80.2% | <0.001 | 75.8% | 79.2% | <0.001 | 77.7% | 77.8% | 1.0 |
| % with HbAS | 20.2% | 18.3% | <0.01 | 20.2% | 19.2% | 0.19 | 11.4% | 19.8% | <0.001 |
| % with HbAC | 1.4% | 1.4% | 0.99 | 1.5% | 1.3% | 0.44 | 0.3% | 1.4% | 0.12 |
| % with HbSS | 1.3% | 0.0% | <0.01 | 2.0% | 0.2% | <0.001 | 10.0% | 0.6% | <0.001 |
| % with HbSC | 0.4% | 0.0% | 0.003 | 0.5% | 0.1% | <0.001 | 0.6% | 0.3% | 0.67 |
| % rural residence | 64.6% | 53.0% | <0.001 | 67.5% | 56.2% | <0.001 | 81.7% | 60.3% | <0.001 |
| % in top two wealth quintiles | 32.9% | 47.9% | <0.001 | 28.2% | 44.3% | <0.001 | 18.6% | 38.2% | <0.001 |
| % with mothers with secondary school or greater† | 37.2% (n=7102) | 50.0%(n=3203) | <0.001 | 33.3% (n=4317) | 46.9% (n=5988) | <0.001 | 23.7% (n=324) | 41.8% (n=9981) | <0.001 * |

The second row summarises the mean age of children with anaemia and no anaemia for three different severity categories, and rows below show the average percentage of children within each anaemia category. Figures were computed without sample weights.
*P values were estimated using the Wilcoxon rank-sum method for age and an equality of proportions test for the other covariates.
†The number of children (N) in each anaemia severity category are reported in the top row and is reported separately for the maternal education category, which has missing values.
HbAA, normal HBB; HbAC, haemoglobin C trait; HbAS, sickle cell trait; HbSC, heterozygous for HbS and HbC; HbSS, sickle cell anemia; RDT, rapid diagnostic test.

mild-to-severe anaemia (tables 1 and 3). Although HbAS was associated with a slight increase in mild or moderate anaemia risk among RDT- children (1.19 and 1.32 for mild-to-severe and moderate-to-severe, respectively, and a non-significant 0.70 for severe anaemia, table 3), HbAS was protective against severe anaemia among RDT+ children relative to HbAA (0.48, 95% CI: 0.31 to 0.72, online supplemental table S1).

### Anaemia attributable to malaria and SCD
We found the fraction of anaemia attributable to malaria to increase with the severity of anaemia, from 12.4% (95% CI: 11.1% to 13.7%) of overall anaemia (adjusted haemoglobin concentration <110 g/L) to 29.6% (95% CI: 29.6% to 31.8%) of moderate-to-severe anaemia (<100 g/L) and to 70.6% (95% CI: 62.7% to 78.5%) of severe anaemia (<70 g/L) (figure 1A). The fraction attributable to SCD was 0.6% (95% CI: 0.4% to 0.9%) of overall anaemia, 1.3% (95% CI: 1.0% to 1.7%) of moderate-to-severe and 10.6% (95% CI: 6.7% to 14.9%) of severe anaemia (figure 1B).

Although RDT positivity is a major risk factor for anaemia at all levels of severity (table 3), malaria might not be the cause of anaemia among all children who were RDT+. Among RDT+ children with anaemia, we estimated that 69.3% (95% CI: 71.8% to 74.2%) would still have been anaemic if they had been RDT−, 46.0% with moderate-to-severe anaemia (95% CI: 43.3 to 48.7) would have had moderate-to-severe anaemia had they been RDT−, and 12.4% with severe anaemia (95% CI: 8.4 to 17.1) would have had severe anaemia even if they were RDT−.

Children with HbSS or HbSC were at high risk of anaemia (table 3), but some of these children would still have anaemia even if they did not have SCD. If these children with SCD had HbAA instead, 60.0% (95% CI: 65.8% to 70.7%) of those who had anaemia would still have anaemia, and 45.2% (95% CI: 39.3% to 52.9%) and 7.9% (95% CI: 5.1% to 11.8%) with moderate-to-severe and severe anaemia would still have those levels of anaemia.

If the information about HBB type is not used by the regression models used for prediction (online supplemental table S2), then the estimated fraction of anaemia attributable to malaria drops from 12.4% to 12.3%, moderate-to-severe anaemia attributable to malaria drops from 29.6% to 29.4% and severe from 69.3 to 68.0%.

### Predictors of haemoglobin concentration
Each year of age was associated with an increase in haemoglobin of 2.74 g/L (table 4). Among children with HbAA, RDT positivity was associated with a 12.93 g/L lower haemoglobin concentration (table 4). Among malaria negative children, haemoglobin concentration was 28.92 g/L lower among children with HbSS than HbAA. However, among RDT+ children, those with HbAS had higher haemoglobin concentration than HbAA (2.17, 95% CI: 0.90 to 3.45 g/L, online supplemental table S3). Each of the lowest two wealth quintiles had lower haemoglobin concentrations than the combined top two

**Table 3** Predictors of anaemia

| | Adjusted haemoglobin <110 g/L (anaemia) | | | Adjusted haemoglobin <100 g/L (moderate-to-severe anaemia) | | | Adjusted haemoglobin <70 g/L (severe anaemia) | | |
|---|---|---|---|---|---|---|---|---|---|
| | OR | (2.5% to 97.5%) | P value | OR | (2.5% to 97.5%) | P value | OR | (2.5% to 97.5%) | P value |
| Age in years | 0.68 | (0.65 to 0.71) | <0.001 | 0.71 | (0.69 to 0.74) | <0.001 | 0.73 | (0.67 to 0.80) | <0.001 |
| Female | 0.86 | (0.77 to 0.96) | <0.01 | 0.84 | (0.76 to 0.93) | <0.001 | 0.87 | (0.67 to 1.14) | 0.31 |
| HBB/malaria status | | | | | | | | | |
| HbAA/− (ref) | -- | | | -- | | | -- | | |
| HbAS/− | 1.19 | (1.00 to 1.41) | 0.04 | 1.32 | (1.13 to 1.55) | <0.001 | 0.70 | (0.28 to 1.76) | 0.45 |
| HbAC/− | 1.29 | (0.77 to 2.17) | 0.34 | 1.43 | (0.86 to 2.38) | 0.17 | * | * | * |
| HbSC/− | 12.05 | (1.77 to 81.82) | 0.01 | 2.43 | (0.58 to 10.14) | 0.22 | * | * | * |
| HbSS/− | 69.66 | (9.63 to 503.87) | <0.001 | 22.93 | (10.56 to 49.78) | <0.001 | 76.14 | (32.71 to 177.26) | <0.001 |
| HbAA/+ | 4.05 | (3.49 to 4.71) | <0.001 | 5.06 | (4.51 to 5.68) | <0.001 | 10.96 | (6.96 to 17.25) | <0.001 |
| HbAS/+ | 3.39 | (2.67 to 4.30) | <0.001 | 4.23 | (3.49 to 5.12) | <0.001 | 5.21 | (2.87 to 9.46) | <0.001 |
| HbAC/+ | 2.90 | (1.61 to 5.23) | <0.001 | 4.95 | (3.03 to 8.10) | <0.001 | 1.56 | (0.21 to 11.58) | 0.66 |
| HbSC/+ | † | † | † | 22.93 | (5.10 to 103.13) | <0.001 | 52.36 | (9.17 to 298.96) | <0.001 |
| HbSS/+ | † | † | † | 62.43 | (13.83 to 281.76) | <0.001 | 170.33 | (64.52 to 449.66) | <0.001 |
| Rural (urban ref) | 0.99 | (0.85 to 1.14) | 0.86 | 0.92 | (0.80 to 1.05) | 0.22 | 1.62 | (1.13 to 2.31) | 0.01 |
| Wealth | | | | | | | | | |
| Richer/richest (ref) | -- | | | -- | | | -- | | |
| Middle | 1.03 | (0.88 to 1.20) | 0.71 | 1.09 | (0.93 to 1.28) | 0.28 | 0.96 | (0.60 to 1.53) | 0.87 |
| Poorer | 1.33 | (1.09 to 1.63) | <0.01 | 1.34 | (1.11 to 1.62) | <0.01 | 1.27 | (0.82 to 1.97) | 0.29 |
| Poorest | 1.72 | (1.36 to 2.14) | <0.001 | 1.51 | (1.24 to 1.84) | <0.001 | 1.56 | (0.97 to 2.49) | 0.06 |
| Mother's education | | | | | | | | | |
| None or primary (ref) | -- | | | -- | | | -- | | |
| Secondary or higher | 0.85 | (0.73 to 1.00) | <0.05 | 0.85 | (0.74 to 0.98) | 0.03 | 0.95 | (0.68 to 1.33) | 0.76 |

Multivariate logistic regression model for predicting mild-to-severe (left columns), moderate-to-severe (middle columns) and severe anaemia (right columns). The effect of risk factors expressed as ORs.
*ORs could not be estimated because no observed children in this category were severely anaemic.
†ORs could not be estimated because all observed children in this category were anaemic.
HbAA, normal HBB; HbAC, haemoglobin C trait; HbAS, sickle cell trait; HbSC, heterozygous for HbS and HbC; HbSS, sickle cell anemia.

and having a mother with a secondary school or higher education had higher haemoglobin concentrations than a mother with primary or no education (1.12, 95% CI: 0.28 to 1.97).

## DISCUSSION

We found that malaria RDT positivity and SCD were associated with a high risk of anaemia among children participating in Nigeria's 2018 DHS. Our results support the hypothesis that malaria and SCD play a larger role in more severe compared with mild anaemia. Our estimate of the contribution of malaria to moderate-to-severe anaemia among young children in Nigeria of 30% is substantially higher than the GBD's estimate of 12% (http://ghdx.healthdata.org/gbd-results-tool). We found that higher levels of household wealth and maternal education were associated with reduced risk of anaemia and higher average haemoglobin concentration in children, which is consistent with earlier analyses of the same dataset[7 12] and more broadly in SSA.[13] However, the effects of malaria

RDT positivity and SCD on anaemia were much larger than these sociodemographic factors.

Testing for malaria infection could help assess the aetiology of anaemia in individuals, but when malaria prevalence is high, many children with anaemia from non-malarial causes could incidentally have asymptomatic or mild malaria. In our analysis, 81% of children with adjusted haemoglobin concentration <70 g/L were malaria-positive, but our model indicates that 12% of these children would have had anaemia even if they were not malaria-positive. At the less stringent <100 g/L threshold, 46% of malaria-positive children with moderate-to-severe anaemia would have had anaemia if they had not been malaria-positive, indicating that malaria is a larger contributor to more severe anaemia. These findings are important because they mean that anaemia in children with malaria cannot be assumed to be due to malaria and therefore the anaemia needs to be investigated and managed appropriately. Previous studies have found that a high proportion of children with apparent symptomatic

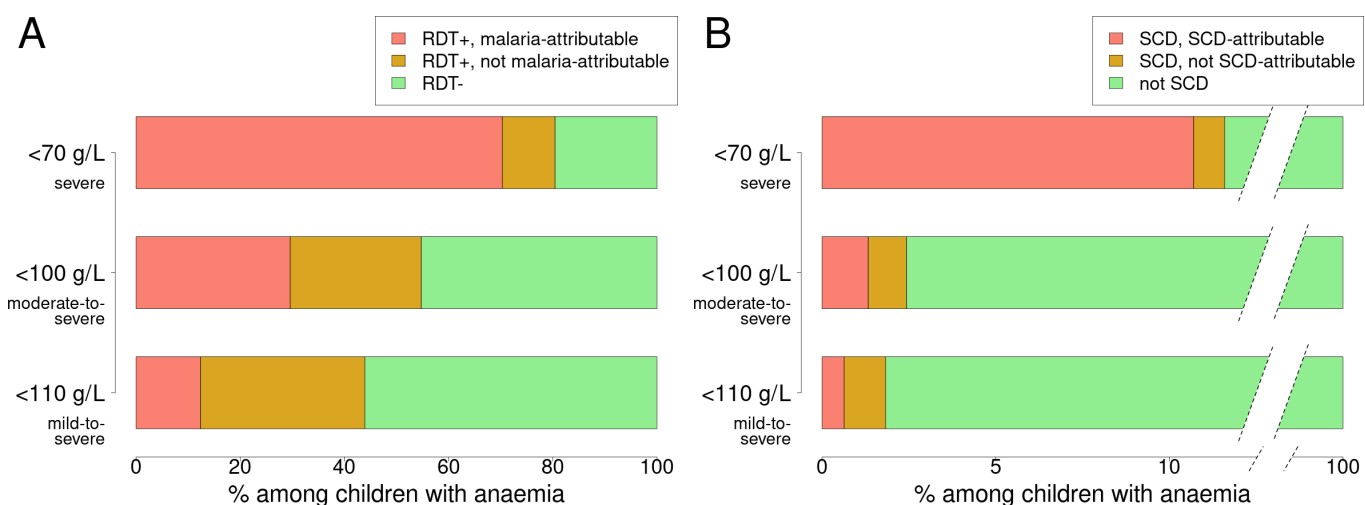

**Figure 1** Fraction of anaemia attributable to malaria or SCD. Anaemia is defined as having an adjusted haemoglobin concentration of <110 g/L (bottom bars), <100 g/L (middle bars) or<70 g/L (top bars). (A) The bars represent the fraction of these three levels of anaemia that can be malaria-attributed (anaemic children who were RDT+ but would not have been anaemic if they had been RDT− according to our model), RDT+ but not malaria-attributed (RDT+ children who would have been anaemic even if they were RDT-), and RDT−. (B) The fraction of anaemic children that can be attributed to SCD. RDT, rapid diagnostic test; SCD, sickle cell disease.

| Table 4 | Predictors of haemoglobin concentration | | |
|---|---|---|---|
| | g/L | (2.5% to 97.5%) | P value |
| Baseline concentration | 99.29 | (98.06 to100.52) | <0.001 |
| Age in years | 2.74 | (2.50 to 2.97) | <0.001 |
| Female (ref: male) | 1.23 | (0.61 to 1.85) | <0.001 |
| HBB/malaria status | | | |
| HbAA/− (ref) | -- | | |
| HbAS/− | −1.87 | (−2.80 to −0.94) | <0.001 |
| HbAC/− | −1.15 | (−3.89 to 1.58) | 0.41 |
| HbSC/− | −12.35 | (−15.90 to −8.80) | <0.001 |
| HbSS/− | −28.92 | (−33.31 to −24.52) | <0.001 |
| HbAA/+ | −12.93 | (−13.76 to −12.10) | <0.001 |
| HbAS/+ | −10.76 | (−12.02 to −9.49) | <0.001 |
| HbAC/+ | −9.42 | (−12.65 to −6.20) | <0.001 |
| HbSC/+ | −27.85 | (−35.65 to −20.06) | <0.001 |
| HbSS/+ | −37.25 | (−43.56 to −30.94) | <0.001 |
| Rural (ref: urban) | −0.03 | (−0.87, 0.81) | 0.94 |
| Wealth | | | |
| Richer/richest (ref) | -- | | |
| Middle | −0.15 | (−1.05 to 0.76) | 0.75 |
| Poorer | −1.89 | (−3.04 to −0.74) | <0.01 |
| Poorest | −3.80 | (−5.10 to −2.50) | <0.001 |
| Mother's education | | | |
| None or primary (ref) | -- | | |
| Secondary or higher | 1.12 | (0.28 to1.97) | <0.01 |

Multivariate linear regression results to predict adjusted haemoglobin concentration. The baseline haemoglobin concentration is in the first row, with the following rows summarising the effect of risk factors. HbAA, normal HBB; HbAC, haemoglobin C trait; HbAS, sickle cell trait; HbSC, heterozygous for HbS and HbC; HbSS, sickle cell anemia.

malaria might have had a different underlying aetiology for their symptoms. An estimated 1/3 of a cohort of children hospitalised for severe malaria in Kenya may have had non-malarial illness.[14][15] estimated that only 37% of fevers among malaria-positive children in SSA were due to malaria.[15]

Although only 1.3% of children had SCD (HbSS or HbSC), our analysis attributes 10.6% of severe anaemia (haemoglobin concentration <70 g/L) to SCD. We note that 11.6% of children with severe anaemia had HbSS or HbSC. Ideally all children with SCD should be identified through a newborn screening programme. However, such programmes are scarce in Africa but as an interim measure our findings indicate that, because SCD is so prevalent in our population-representative sample of children with severe anaemia, screening all young children with severe anaemia with a point-of-care test could identify undiagnosed SCD. Active management could then be started, which would improve their intellectual and physical development and could reduce their need for transfusions.[16] describe a risk-assessment algorithm that identified 73% of undiagnosed sickle cell anaemia among children admitted to the hospital for severe anaemia.[16]

Although sickle trait was associated with an increased risk of mild anaemia, it was associated with protection against anaemia among malaria-positive children, with a greater degree of protection against severe anaemia. This is consistent with observations that sickle trait protects against severe malaria more than mild malaria illness[4] and reduces hospital admissions for malaria (OR=0.26).[17] Sickle trait did not appear to be associated with protection against malaria infection since children with HbAS did not have lower malaria RDT positivity (online supplemental figure S2).[18] Our estimate of the protection of

HbAS against malaria-associated anaemia is generally consistent with the literature. Sickle cell trait may confer over 90% protection against severe malarial anaemia.[4 19] Lopera-Mesa et al found a clinical malaria incidence rate ratio of 0.66 between children with HbAS versus HbAA, with protection peaking around 3–6 years of age[20]

Our study has several limitations. Although we found that increasing age was associated with a reduced risk of anaemia, this finding could be partially attributed to the fixed thresholds used to define anaemia for all children regardless of age. Haemoglobin concentrations increased with age, which has been observed in healthy, disease-free children.[21] Using a single threshold for all ages could result in an overestimate of anaemia among young children and an underestimate at older ages. Of note, current anaemia thresholds were set in 1968 using mostly European and North American populations, and the WHO has begun a process to revise them.[22] The DHS interviews were conducted between August and December, which overlap with malaria season in northern Nigeria, and the estimates of malaria prevalence and its impact on anaemia could be different across seasons. We estimated the fraction of anaemia attributable to malaria and SCD using a simple statistical model, which would have been more accurate if assays for other major causes of anaemia, such as helminths, genetic conditions besides SCD and iron or other micronutrient deficiency, were included. A combination of these tests would be key to determining the aetiology of an individual's anaemia. We equated malaria RDT status with malaria, but the antigens detected by the RDT can persist for weeks after malaria treatment.[23] In addition, the sensitivity of the RDT can be affected by the subject's age, sex and urban versus rural residence.[24] Similarly, only data on P. falciparum malaria were available in the DHS, not on P. vivax, though falciparum comprises most infections is in Nigeria. Even with additional data, the causes of anaemia are complex and can be interrelated, making it difficult to assign aetiology. Calis et al 2008 used structural equation modelling to account for these relationships.[25] We were limited to data available in the DHS, and our simpler approach was able to capture the relationship between malaria positivity, sickle trait and SCD. We found that using the malaria test data alone (ie, ignoring the HBB type) would slightly underestimate the fraction of anaemia attributable to malaria. Keeping this effect in mind, we believe that our approach can be applied to DHS datasets that do not include HBB typing to estimate the fraction of anaemia attributable to malaria.

Our analyses highlight the importance of malaria and SCD in the more severe stages of anaemia among young children in Nigeria. The majority of children in the DHS survey had mild-to-severe anaemia, most of which was not due to malaria or SCD. At the more severe end of the anaemia spectrum, malaria appears to be linked to 70.6% of severe cases and SCD to 10.6%. There are many ways to prevent or mitigate anaemia early, which could avert the development of severe anaemia and the need for blood transfusions. Malaria control has been observed to reduce anaemia prevalence.[26 27] Mass iron supplementation or fortification is suggested for regions with high anaemia prevalence, but because iron supplementation can increase the susceptibility of malaria, malaria diagnosis, prevention, and treatment are important components of anaemia-reduction in malaria-endemic regions.[28] However, existing coverage of such antimalaria measures is often inadequate for the safe provision of iron to children.[27] Abhulimhen-Iyoha et al 2018 found that the majority of blood transfusions required by children in a tertiary hospital in Nigeria were due to severe malaria and suggest that malaria prevention could alleviate pressure on the nation's limited blood supply.[29] The early diagnosis and prophylactic treatment for SCD in certain populations could also be part of an anaemia-control strategy, a possibility made more feasible with the recent development of point-of-care tests for the condition.[30] We believe that a more complete and region-specific understanding of the aetiology of anaemia, particularly severe anaemia, is required to prioritise prevention efforts and assist diagnosis and appropriate treatment.

**Author affiliations**
[1]Institute for Disease Modeling, Bill & Melinda Gates Foundation, Seattle, Washington, USA
[2]Institute for Health Metrics and Evaluation, University of Washington, Seattle, Washington, USA
[3]Department of Public Health, Federal Ministry of Health, Abuja, Nigeria
[4]Department of Paediatrics, University of Abuja College of Health Sciences, Abuja, Nigeria
[5]Department of International Public Health, Liverpool School of Tropical Medicine, Liverpool, UK
[6]Department of Epidemiology and Biostatistics, Imperial College London, London, UK
[7]Centre of Excellence for Sickle Cell Disease Research & Training (CESRTA), University of Abuja, Abuja, Nigeria

**Acknowledgements** ON is the primary investigator of the Sickle Pan African Research Consortium in Nigeria and acknowledges support from the National Heart, Lung, and Blood Institute for the Sickle Pan African Research Consortium. FBP acknowledges infrastructure support for the Department of Epidemiology and Biostatistics provided by the NIHR Imperial Biomedical Research Centre. We thank the Demographic and Health Survey programme and the National Population Commission and the Department of Health, Non-Communicable Diseases Unit, Federal Ministry of Health, Nigeria for making our work possible. We also thank Michelle O'Brien and Edward Wenger for helpful conversations.

**Contributors** DLC, APO, GC-C and ON conceived the study. DLC and APO designed the study. DLC and APO performed the analyses and verified underlying data. DLC, APO, GC-C, AS, UN-A, IB, FBP and ON interpreted the data and results. DLC is the guarantor of this work. All authors participated in the writing of the manuscript.

**Funding** OEN received support through the Sickle Pan African Research Consortium NigEria Network (SPARC-NEt) (NIH U01HL156942). Nigeria's DHS was funded by the United States Agency for International Development, the Global Fund, the Bill & Melinda Gates Foundation, the United Nations Population Fund, and the World Health Organisation.

**Competing interests** FBP received fees from Bluebird Bio, Novartis, and Analytics Group for work unrelated to this study.

**Patient and public involvement** Patients and/or the public were not involved in the design, or conduct, or reporting or dissemination plans of this research.

**Patient consent for publication** Not applicable.

**Ethics approval** This study uses only publicly available DHS data. The DHS survey was approved by the National Health Research Ethics Committee of Nigeria (NHREC) and the ICF Institutional Review Board.

**Provenance and peer review** Not commissioned; externally peer reviewed.

**Data availability statement** The data used in this study can be obtained from the Demographic and Health Survey programme by registered users upon approval of a study description. Instructions for obtaining these data are at https://dhsprogram.com/.

**ORCID iDs**
Dennis L Chao http://orcid.org/0000-0002-8253-6321
Uche Nnebe-Agumadu http://orcid.org/0000-0002-0101-5605
Imelda Bates http://orcid.org/0000-0002-0862-8199
Frédéric B Piel http://orcid.org/0000-0001-8131-7728
Obiageli Nnodu http://orcid.org/0000-0002-4801-3156

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
