## [Reviewer comments · BMJ Open]

ARTICLE DETAILS

TITLE (PROVISIONAL)	The contribution of malaria and sickle cell disease to anaemia among children aged 6 to 59 months in Nigeria: A cross-sectional study using data from the 2018 Demographic and Health Survey
AUTHORS	Chao, Dennis; Oron, Assaf; Chabot-Couture, Guillaume; Sopekan, Alayo; Nnebe-Agumadu, Uche; Bates, Imelda; PIEL, FRÉDÉRIC B.; Nnodu, obiageli

VERSION 1 – REVIEW

REVIEWER	Buchwald, Andrea University of Maryland School of Medicine, Pediatrics
REVIEW RETURNED	25-May-2022

GENERAL COMMENTS	This manuscript describes the relationship between sickle cell disease, malaria, and anemia in a cross-sectional study of Nigerian children under the age of 5. There is a rich dataset underlying their analyses and the use of attributable fractions is a strength of this paper. However, much of the data is not presented clearly which makes interpretation slightly difficult. General comment – The survey used only RDT to measure malaria. Older children in particular may carry parasites below RDT detectable thresholds for some time, potentially leading to misclassification of some RDT- individuals as malaria negative. The effect of this should be mentioned in the discussion and authors should be more careful to distinguish between RDT negative and malaria negative. Specific comments: Table 1 – Is the final column numbers or percents? Why is it formatted differently from the other columns? Usually I would expect to see the total column at the end or beginning. Results: Line 27: “Severe anaemia was more prevalent among younger children except for those with HbSS” – there is no statistical test to support this and visually all lines except that for HbSS appear flat. Table 2 – This table is a little disappointing given the rich breakdown in Table 1, why are you only looking at moderate to severe anemia here when using so many other categories in all other analysis? What about HbAA and other categories? What are these p-values from? Chi-square tests? Comparison groups are not obvious for all categories (i.e. maternal education). Please include the N with the percents and include SD with means. Table 3 – When you have odds ratios of greater than 1000 it indicates there is a problem with your model – this is clearly a result of having no children in the not anemic category – the model is not valid here as it requires positivity (a non-zero probability of having an outcome level) so you can’t include results from it.
--

	Table 3 - I think it would be more interesting to use the comparison group of HbSS negative for the models for HbSS – this would give you a separate odds ratio for the effect of having sickle cell (HbSS negative compared to HbAA negative) and for the effect of having malaria among those with sickle cell (HbSS positive compare to HbSS negative) – and do the same for all comparisons. Page 6, lines 48-51 – in order to say that HbAS was protective you need to provide a statistical comparison, just looking at and comparing the odds ratios is insufficient. Same problem with page 7, lines 23-25 and page 7 lines 40-41, page 8, line 37-38 - If this is one of your main findings, please do some actual statistical tests – statistical software should easily produce odds ratios with specific comparisons of interest Page 7, lines 30-32 – this would seem to suggest that SS trait is not important for determining the influence of malaria on anemia, again making a change in references for tables 3 and 4 seem useful
--	---

REVIEWER	Chunda, Catherine University Teaching Hospitals - Lusaka Children's Hospital, Paediatric Haematology/Infectious Diseases
REVIEW RETURNED	31-May-2022

GENERAL COMMENTS	I have no comments to the reviewer
------------------------------------

VERSION 1 – AUTHOR RESPONSE

Reviewer: 1

Dr. Andrea Buchwald, University of Maryland School of Medicine

Comments to the Author:

This manuscript describes the relationship between sickle cell disease, malaria, and anemia in a cross-sectional study of Nigerian children under the age of 5. There is a rich dataset underlying their analyses and the use of attributable fractions is a strength of this paper. However, much of the data is not presented clearly which makes interpretation slightly difficult.

We decided to remove all results related to “semi-severe” anemia, which is not a standard classification and made the tables and text confusing.

General comment – The survey used only RDT to measure malaria. Older children in particular may carry parasites below RDT detectable thresholds for some time, potentially leading to misclassification of some RDT- individuals as malaria negative. The effect of this should be mentioned in the discussion and authors should be more careful to distinguish between RDT negative and malaria negative.

Thank you for this observation. We added a reference to the differences in sensitivity by age and region and added a new point to the “strengths and limitations” section.

Specific comments:

Table 1 – Is the final column numbers or percents? Why is it formatted differently from the other columns? Usually I would expect to see the total column at the end or beginning.

We eliminated this confusing column on “semi-severe” anemia. In the new table, the final column is the total number of children per row.

Results: Line 27: “Severe anaemia was more prevalent among younger children except for those

with HbSS” – there is no statistical test to support this and visually all lines except that for HbSS appear flat.

We agree. The important feature is the sharp increase in severe anemia with age among those with HbSS, and we changed the text to reflect this.

Table 2 – This table is a little disappointing given the rich breakdown in Table 1, why are you only looking at moderate to severe anemia here when using so many other categories in all other analysis? What about HbAA and other categories? What are these p-values from? Chi-square tests? Comparison groups are not obvious for all categories (i.e. maternal education). Please include the N with the percents and include SD with means.

We added comparisons for mild and severe anemia to Table 3. We now describe the source of p-values in the caption. We clarified some of the wording, and added N for the maternal education category, which is missing some data so has a lower N.

Table 3 – When you have odds ratios of greater than 1000 it indicates there is a problem with your model – this is clearly a result of having no children in the not anemic category – the model is not valid here as it requires positivity (a non-zero probability of having an outcome level) so you can't include results from it.

You are correct that having no children in the “not anemic” category prevents us from estimating these odds ratios. We found that the other odds ratios in the same model were not changed when these problematic categories were excluded (we re-fit the model excluding malaria-positive children with HbSS or HbSC). We changed the last column of Table 3 to summarize the odds of severe anemia (instead of semi-severe), and we encountered the same issue because there are no severely anemic children who are malaria-negative with HbAC or HbSC. We added material describing this to the Methods.

Table 3 - I think it would be more interesting to use the comparison group of HbSS negative for the models for HbSS – this would give you a separate odds ratio for the effect of having sickle cell (HbSS negative compared to HbAA negative) and for the effect of having malaria among those with sickle cell (HbSS positive compare to HbSS negative) – and do the same for all comparisons.

HbSS/+ has a small number of subjects (24.5) and all of them have at least mild anemia, so we would not be able to fit the model for mild-to-severe anemia using this reference group. We chose HbAA/- (with 5564 children it is the largest category) as the reference group.

Page 6, lines 48-51 – in order to say that HbAS was protective you need to provide a statistical comparison, just looking at and comparing the odds ratios is insufficient. Same problem with page 7, lines 23-25 and page 7 lines 40-41, page 8, line 37-38

- If this is one of your main findings, please do some actual statistical tests – statistical software should easily produce odds ratios with specific comparisons of interest

Thank you for keeping us honest here. We now use RDT-positive children as the reference to compute the protection of HbAS among RDT-positive children (page 6), using RDT-positive children with HbAA as a reference. We added two supplemental tables to summarize these results. We make a note of this in Methods. This made redundant the paragraph where we estimated the anemia prevalence if children with HbAS had HbAA instead, so we removed it.

Page 7, lines 30-32 – this would seem to suggest that SS trait is not important for determining the influence of malaria on anemia, again making a change in references for tables 3 and 4 seem useful. We added two supplemental tables that replicate Tables 3 and 4 but use RDT+ children as the reference.

Reviewer: 2

Dr. Catherine Chunda, University Teaching Hospitals - Lusaka Children's Hospital

Comments to the Author:
I have no comments to the reviewer

VERSION 2 – REVIEW

REVIEWER	Buchwald, Andrea University of Maryland School of Medicine, Pediatrics
REVIEW RETURNED	08-Sep-2022
GENERAL COMMENTS	All my previous comments have been addressed.